# Sorption Isotherms and Thermodynamic Properties of Pomegranate Peels

**DOI:** 10.3390/foods11142009

**Published:** 2022-07-07

**Authors:** Nihel Ben Slimane, Mohamed Bagane, Antonio Mulet, Juan A. Carcel

**Affiliations:** 1Applied Thermodynamics Research Laboratory, National Engineering School of Gabes, ENIG, University of Gabes, V423+CVP, Gabes 6029, Tunisia; nihel.benslimane@gmail.com (N.B.S.); drmbag1420@yahoo.fr (M.B.); 2Group of Analysis and Simulation of Agri-Food Processes, Food Technology Department, Universitat Politècnica de València, 46022 Valencia, Spain; amulet@tal.upv.es

**Keywords:** water activity, moisture content, GAB model, isosteric heat, enthalpy–entropy compensation theory

## Abstract

Convective drying is the most widely used technique to stabilize by-products in the food industry, permitting later processing. A thorough knowledge of the relationship between moisture content and water activity allows the optimization of not only drying operations, but the settings of storage conditions. Thus, the thermodynamic properties of pomegranate peels were determined during the desorption process. Experimental sorption isotherms at 20, 30, 40 and 50 °C showed type II Brunauer behavior. Eight different theoretical and empirical equations were fitted to the experimental results; the theoretical GAB model and the empirical Peleg model were the ones that achieved the best fit (R^2^ of 0.9554 and 0.974, respectively). The Clausius–Clapeyron equation and the enthalpy–entropy compensation theory were used to determine the thermodynamic parameters. The isosteric heat determined from the sorption isotherms decreased regularly as the equilibrium moisture content rose (from 8423.9 J/mol at 0.11 kgH_2_O/kg d.m. to 3837.7 J/mol at 0.2 kgH_2_O/kg d.m.). A linear compensation was observed between enthalpy and entropy, which indicated an enthalpy-controlled sorption process.

## 1. Introduction

The pomegranate (*Punica granatium* L.) is a fruit native to India and Iran, whose production has spread throughout the entire Mediterranean region and south western America [1]. Of the countries in the Mediterranean basin, Tunisia is one of the largest producers of pomegranates. The crop is mainly concentrated in the region of Gabes, in the south of the country, and the production in this region accounts for around 40% of the total production [2].

Pomegranate peel is the main by-product of the industrialization of this fruit, making up approximately 30–40% of the raw material weight. Thus, its disposal represents a great economic and environmental problem for industries, and it explains why there is so much interest in the identification of alternative uses [3,4]. In this sense, pomegranate peel is considered as an ‘eco-friendly’ source of phenolic compounds [5]. Recent studies have reported the use of pomegranate peel extracts in the medical field because of its therapeutic properties. The great presence of antioxidant compounds, mainly bioactive phenolic components, could play an important role in protecting human health [6,7,8].

However, independent of the increase in value process, these by-products have to be stabilized to prevent degradation reactions. In this sense, convective drying is one the most common food preservation methods. This drying technique is characterized by its simplicity and reliability [9]. The main purpose of drying is to obtain products with reduced water activity (a_w_), which limits the undesirable degradation of interesting compounds, thereby prolonging shelf life. The moisture content reduction also makes the later transport and storage operations easier. In addition, convective drying can induce other physical, chemical and biological changes that can affect the final quality of the product [10,11]. Most of these reactions are controlled by the relationship between the moisture content and water activity. Therefore, it is necessary to have a thorough knowledge of the sorption isotherms of this by-product. Thus, an accurate modelling of the sorption isotherms permits to establish the best storage conditions or the estimation of the equilibrium moisture content of the sample in the air drying conditions used and allows for the determination of both the optimum end point of drying, which preserves product quality and saves energy [12,13]. In fact, numerous scientific articles can be found that focus on the estimation of the sorption isotherms of very different agricultural products, such as chia, pepper, cambucci or pumpkin [14,15,16,17]. The experimental sorption isotherms can be determined using several methods, and the hygrometric method is one of the most commonly applied due to its simplicity and low cost. Moreover, it provides accurate values of the relationship between water activity and moisture content [18,19,20] and permits the hydrophilic/hydrophobic character of the considered product to be addressed [21].

To our knowledge, there are no reported sorption isotherms and thermodynamic properties of pomegranate peels. Therefore, the aim of this study is to assess the desorption isotherms of pomegranate peels at different temperatures and to determine enthalpy, entropy and Gibbs free energy. The theory of enthalpy–entropy compensation is also applied.

## 2. Materials and Methods

### 2.1. Raw Material

Fresh pomegranates were purchased in a local market (Gabes, Tunisia) and stored at 4 °C until use. The peels were separated from the fruit by hand and immediately used for the experimental determination of the sorption isotherm. The average initial moisture content of the pomegranate peels was measured by differential weighing after keeping the pomegranate peel samples in a vacuum oven at 70 °C until reaching a constant weight (24 h approx.).

### 2.2. Experimental Determination of Sorption Isotherms

The desorption isotherms of pomegranate peels were experimentally determined in a temperature range from 20 to 50 °C by using the hygrometric method. For this purpose, peel pomegranate samples were placed into a sample holder and then into a laboratory convective oven at 60 °C. This temperature is widely used for the drying of food products and by-products because it represents a compromise solution between drying kinetics and the preservation of interesting compounds [22]. At different pre-set times, the samples were extracted and both water activity and moisture content were measured. The water activity was determined for each drying time with a standardized conductivity hygrometer Novasina TH-500 (Air Systems for Air Treatment, Pfaffikon, Switzerland). The hygrometer was previously calibrated using saturated solutions of different salts, which covered the whole range of water activity (LiCl, MgCl_2_, Mg(NO_3_)_2_, NaCl, BaCl_2_ and K_2_Cr_2_O_7_). This procedure provided a rapid and reliable means of measuring water activities, avoiding the degradation of samples, such as when it takes a long time to reach equilibrium. The water activity was measured in triplicate in each sample at 20, 30, 40 and 50 °C. Once water activity was measured, the sample moisture content was determined, also in triplicate, using the AOAC method [23]. The determination was carried out at 70 °C and 800 mbar vacuum levels until samples reached constant weight [21,24].

This procedure was replicated at different drying times until enough data were obtained to be able to describe the whole range of water activity, from 0 to 1.

### 2.3. Modeling

Different models were considered to describe the desorption isotherms of pomegranate peel: two theoretical models, GAB (Guggenheim–Anderson–Boer) and BET (Brunauer–Emmett–Teller), and six semi-empirical models, Caurie, Freundlich, Henderson, Oswin, Peleg and Smith. All these models have been widely used to describe sorption isotherm curves of agricultural materials [25]. Table 1 summarizes the equations of each and the meaning of their respective parameters.

To identify these model parameters, the least squares method was applied by using MATLAB software (The Mathworks, Inc, Natick, MA, USA, License123456, 2016 version). The goodness of fit was evaluated through the determination coefficient (R^2^, Equation (1)), the mean squared error (RMSE, Equation (2)) and the sum of the squared error (SSE, Equation (3)):(1)$$R^{2}=\frac{\sum_{i=1}^{N}(X_{\mathrm{pre},i}-{\bar{X_{\exp,i})}}^{2}}{\sum_{i=1}^{N}{\left(X_{\exp,i}-X_{\mathrm{pre},i}\right)}^{2}}$$
(2)$$\mathrm{RMSE}=\sqrt{\frac{\sum_{i=1}^{N}{\left(X_{\exp,i-}X_{\mathrm{pre},i}\right)}^{2}\;}{N}}$$
(3)$$\mathrm{SSE}=\frac{1}{N}{\left(X_{\exp,i}-X_{\mathrm{pre},i}\right)}^{2}$$
where N is the number of the samples; X_exp,i_ and X_pre,i_ are the experimental and predicted equilibrium moistures, respectively; and $\bar{X_{exp,i}}$ is the average value of the experimental equilibrium moisture.

### 2.4. Determination of Isosteric Heat

In desorption processes, isosteric heat corresponds to the energy required to break the bonds between the absorbent surface and the vapor molecules [27]. Thus, the total isosteric heat of sorption (Q_st_) is the sum of the net isosteric heat of the sorption (q_stn_) and the enthalpy of the vaporization of pure water (ΔH_H2O_), as shown in Equation (4).
(4)$$Q_{\mathrm{st}}=q_{\mathrm{stn}}+\Delta H_{H_{2}O}$$
q_stn_ (J/mol) can be calculated by applying the Clausisus–Clapeyron equation (Equation (5)):(5)$${\left[\frac{\partial ({\mathrm{Lna}}_{w)}}{\partial \left(\frac{1}{T}\right)}\right]}_{X_{w}}=-\frac{Q_{\mathrm{st}}-\Delta H_{H_{2}O}\;}{R}=-\frac{q_{\mathrm{stn}}}{R}$$
where T is the absolute temperature (K) and R is the universal gas constant (8.31 J/mol·K). The experimental data of a_w_ obtained at different temperatures for a given equilibrium moisture content were used in the fit of Equation (5). Thus, q_stn_ was obtained from the slope of the relationship of Ln(a_w_) vs. 1/T.

### 2.5. Differential Sorption Enthalpy and Sorption Entropy

The differential enthalpy (∆H) indicates the state of the water in a biological material depending on the type of force between the water vapor and the sorption sites [27]. However, the differential entropy (∆S) is the relationship between the number of sorption sites with a certain level of power inherent in biological material (Equation (6)) [27]. Gibbs free energy (∆G) describes the spontaneity of the process and it is defined by Equation (7) [27]:(6)$$\Delta S=\frac{\Delta H-\Delta G}{T}$$
(7)$$\Delta G=\mathrm{RT}\;\mathrm{Ln}\left(a_{w}\right)$$

By combining Equations (6) and (7), the differential sorption enthalpy and entropy can be estimated by Equation (8), with the slope of this linear relationship being ∆H/R and the intercept to the Y-axis, ∆S/R [27].
(8)$$\mathrm{Ln}\left(a_{w}\right)=\frac{\Delta S}{R}-\frac{\Delta H}{\mathrm{RT}}$$

### 2.6. Enthalpy–Entropy Compensation Theory

Enthalpy–entropy compensation theory involves the physical and chemical phenomena produced in water sorption processes. This theory proposes a linear relationship between ∆H and ∆S for the sorption of water, which is given by (Equation (9)).
(9)$$\Delta H=T_{B}\Delta S+\Delta G_{B}$$
where T_B_ is the isokinetic temperature (K), namely, the temperature at which all reactions take place at the same speed; and ∆G_B_ is the Gibb free energy (J/mol) at T_B_ [28,30].

Identifying the values of T_B_ and ∆G_B_, the theory was tested by comparing the isokinetic temperature, T_B_, and the harmonic temperature, T_hm_ (Equation (10)). The theory is accepted only if T_B_ ≠ T_hm_ [27].
(10)$$T_{\mathrm{hm}}=\frac{N}{\sum_{1}^{N}(\frac{1}{T})}$$
where N corresponds to the total number of isotherms.

## 3. Results and Discussion

### 3.1. Experimental Moisture Sorption Isotherms

The experimental sorption isotherms of pomegranate peels obtained at 20, 30, 40 and 50 °C are shown in Figure 1. The experimental data of the water activity studied ranged from 0.045 to 0.992. With regard to the moisture content, the values varied from 0.02 to 2.61 kg H_2_O/kg dm. The curves followed the sigmoid shape of a type II isotherm, according to Brunauer’s classification [31]. Thus, three main different regions can be distinguished in the type II isotherms. The first one, corresponding to water activity (a_w_) values of under 0.3, is related to the moisture of the monolayer. The second region, where aw ranged from 0.3 to 0.7, corresponds to the organization of water molecules in several layers. In this region, the relationship between water activity and moisture content is linear. Finally, in the third region, which corresponds to aw values above 0.7, there are water molecules that are weakly linked to the product matrix and, therefore, are available for chemical and microbiological reactions [30].

Over the range studied, the temperature had a slight, but significant, effect on the desorption isotherms of the pomegranate peels. Similar results were found by Rosa et al. [30] when studying papaya seeds. This phenomenon could be attributed to the state of excitation of the molecules: the higher the temperature, the more excited the molecules. Thus, when the water content increases and, therefore, the forces of attraction between water molecules and the product matrix decrease, the influence of temperature could become more significant [28].

### 3.2. Modeling of the Sorption Isotherms of the Pomegranate Peels

Different models were fitted to the experimental data in order to identify the best fit for the mathematical description of the isotherms. The goodness of the fit was assessed according to the correlation coefficient (R^2^), the root-mean-square error (RMSE) and the sum of the squared error (SSE), as explained in the Materials and Methods Section. As can be observed in Table 2, the theoretical GAB and BET models and the semi-empirical Peleg model were those that provided the best fit. Thus, these models achieved an R^2^ value of over 0.95 and RMSE and SSE of under 1, at every temperature tested. These results were similar to those reported by Rosa et al. [30] and Paes et al. [16] for papaya seeds and cambucci, respectively. Moreover, these three models fitted adequately the whole aw–moisture content range under study (Figure 1).

As for the identified parameters of the Peleg model, all of them showed a tendency to increase as the temperature rose, confirming the influence of this parameter. Kammoun-Bejar et al. [32] found similar results for orange peels. Other authors also reported that the Peleg model offered a good fit to other more dissimilar agricultural products, such as pepper, papaya or fig [15,30,33].

As for the theoretical GAB model, the values of the identified model parameters (Table 2) mean that the pomegranate peel isotherm can be classified in the category of 0 < K ≤ 1, C ≥ 2 [34]. As regards the effect of the temperature, it was observed that the monolayer moisture content (X_m_) was greater as the temperature decreased. This parameter is linked to the energy interactions of water molecules in the monolayer in a specific sorption site of the material. The reduction in the moisture content of this monolayer that occurs as the temperature rises may be related to a reduction in the total number of active sites for water bonding, because of physical and/or chemical modifications [30]. As concerns the K parameter, the value was almost constant and close to 1, which indicates that the GAB equation became the BET model [35]. As to the C parameter, it ranged from 4 ± 0.8 at 20 °C to 12 ± 1 at 50 °C. The higher the C parameter, the more tightly the water is bound in the monolayer [36].

### 3.3. Isosteric Heat of Sorption

At a specific moisture content of the sample, the desorption heat was determined using the Clausius–Clapeyron equation (Equation (5)). The values of q_stn_ were estimated from the slope of the relationship between ln (a_w_) vs. (1/T). This is shown in Figure 2 for equilibrium moisture contents varying between 0.11 and 0.2 kgH_2_O/kg dm. As can be observed in Table 3, the q_stn_ decreased from 8423.9 J/mol for 0.11 kgH_2_O/kg dm to 3837.7 J/mol for 0.2 kgH_2_O/kg dm.

Table 3 shows how both the net isosteric heat of sorption (q_stn_) and the total isosteric heat of sorption (Q_st_) evolve in line with the moisture content. This can be used for the estimation of the binding energy and for the purposes of assessing the availability of polar sites for water vapor. Bougayr et al. [28] found similar results when analyzing sewage sludge and Kammoun Bejar et al. [32] studying orange peels. The net isosteric heat of sorption (q_stn_) showed a close relationship with the moisture content. It decreased as the equilibrium moisture content increased. Thus, the maximum isosteric heat of sorption was obtained at the lowest equilibrium moisture content. This would indicate that the bond between the water molecules and the pomegranate peel matrix first occurred in the more active sorption sites, which generated strong interactions [35]. As for Q_st_, the values obtained ranged from 51,923.9 J/mol for 0.11 kg H_2_O/kg dm to 47,337.7 J/mol for 0.2 kg H_2_O/kg dm (Table 3). The values of Q_st_ tended towards those of pure water vaporization enthalpy (ΔHwatervapor), as X_w_ increased [27]. The positive values of the net isosteric heat illustrated the endothermic nature of the process of desorption [30].

### 3.4. Thermodynamic Properties

The differential enthalpy and entropy (ΔH and ΔS) permit the evaluation of the thermodynamic behavior of water in sorption. Thus, Figure 3 and Figure 4 show how these pomegranate peel properties evolve in line with the moisture content. As can be observed, the ΔH and ΔS values decreased as the equilibrium moisture content rose. Other authors have found a similar trend for different agricultural products [33,37,38]. The ΔS represents the variation of the total entropy of water produced when new water molecules are absorbed into the food matrix. In the case of the pomegranate peel, ΔS showed its maximum value at the lowest moisture content. In these conditions, the sorption of water occurs where the most active sites are found, that is, in the monolayer. Thus, the water molecules are less readily available for deterioration reactions due to the fact that they are less mobile as a consequence of the greater polarity at the surface [27].

Figure 3 shows that the differential enthalpy increased when the moisture content decreased, reaching to the maximum value at the lowest moisture content. This can be attributed to changes in the bonding forces between the water and the product. At the beginning of sorption, which occurs at very low moisture content, there are active polar sorption sites, which are available to be covered by water molecules. When water molecules occupy these sorption sites, the sorption itself begins to take place at less active sites. This means a reduced interaction energy and, therefore, lower differential enthalpy [20].

### 3.5. Enthalpy–Entropy Compensation Theory

As can be observed in Figure 5, a linear relationship was identified (R^2^ = 0.970) between the ΔH and ΔS values obtained at different equilibrium moisture contents. The isokinetic temperature (T_B_) and the harmonic mean temperature (T_hm_) were identified and compared following Equation (10). Thus, the T_hm_ value obtained, 307.7 K, was significantly different from the T_B_ value, 675.5 K. This indicated that the sorption phenomena in pomegranate peels agreed the compensation theory. Thus, according to Hssaini et al. [33], the fact of T_B_ was greater than T_hm_ indicated that the process is enthalpy driven and then the microstructure of pomegranate remained stable during moisture sorption [30,33]. The free energy, ∆G_B_, can be used as an indicator of the affinity of the product for water. Its positive value, 4136.59 J/mol, indicated that the sorption was a non-spontaneous process [30,33].

## 4. Conclusions

The sorption isotherms for pomegranate peels presented a type II shape according to the Brunauer classification. The GAB, BET and Peleg models provided the best fit for the experimental data, being useful tools for the purposes of describing the sorption isotherms of pomegranate peels. The temperature affected the experimental sorption process. This influence was also observed in the evolution with the temperature of the models’ parameters identified. Thus, the monolayer moisture content (X_m_ of GAB model) decreased as the temperature raised. The net isosteric heat of sorption was greater at a lower moisture content, providing interesting data with which to predict both the energy requirement in pomegranate peel dehydration processes and the behavior of water molecules in sorption. The sorption of water in pomegranate peels was controlled by the enthalpy mechanisms that were non-spontaneous. This study is a prior step to addressing pomegranate drying and defining the best storage conditions

## Figures and Tables

**Figure 1 foods-11-02009-f001:**
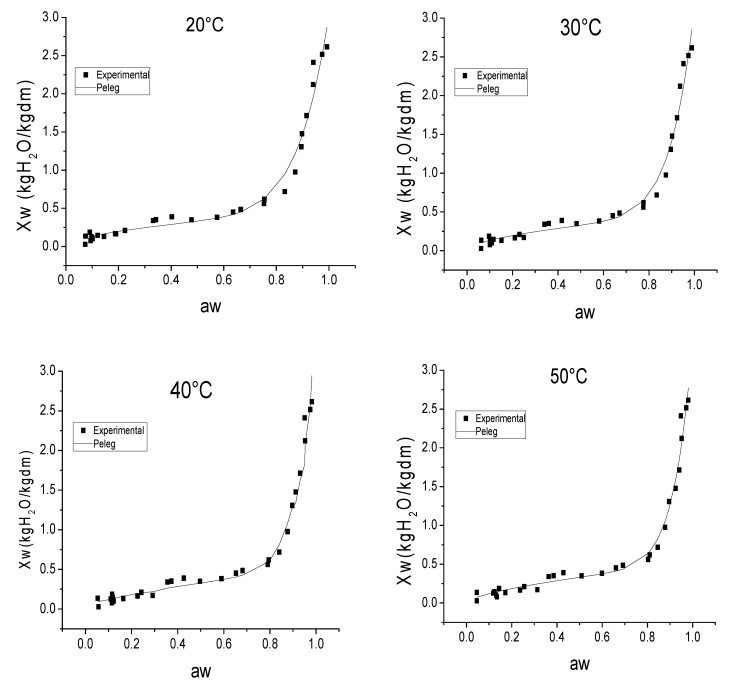
Desorption isotherms of pomegranate peels at 20, 30, 40 and 50 °C, both experimental and those calculated using the Peleg equation. Each experimental point represents the average of, at least, three replicates.

**Figure 2 foods-11-02009-f002:**
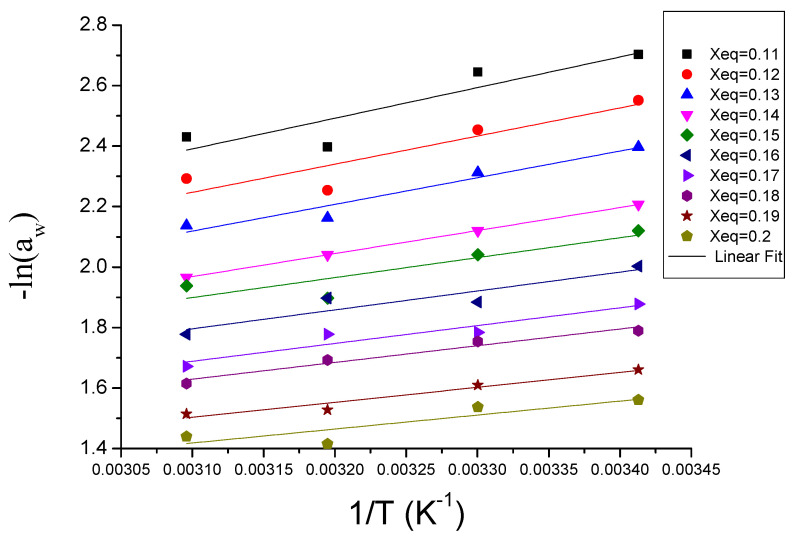
Relationship between the natural logarithm of water activity (ln(a_w_)) and the reverse of the temperature at different moisture contents (kg H_2_O/kg d·m) of pomegranate peels.

**Figure 3 foods-11-02009-f003:**
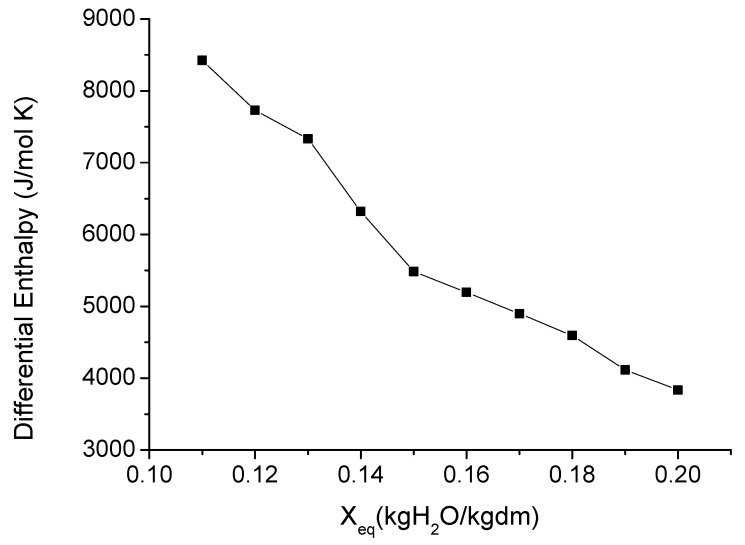
Evolution of the differential enthalpy of pomegranate peels in line with the moisture content.

**Figure 4 foods-11-02009-f004:**
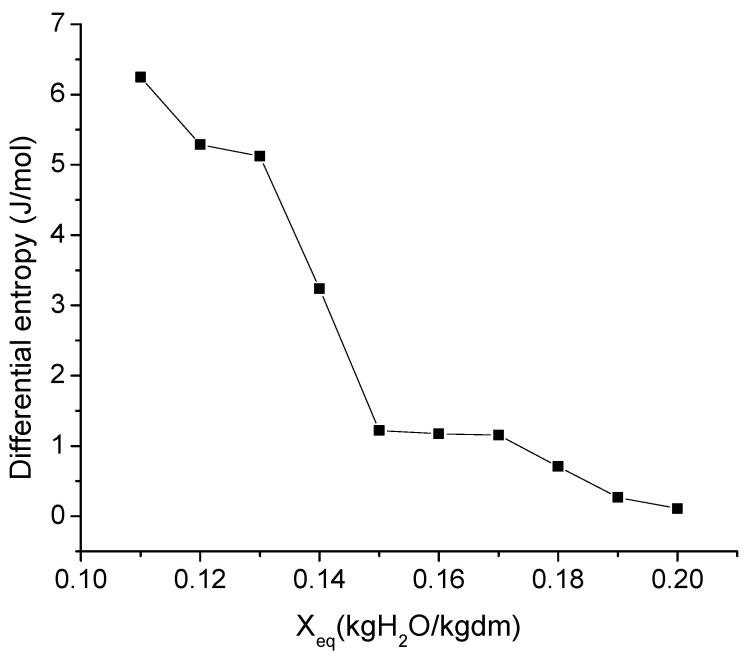
Evolution of the differential entropy of pomegranate peels in line with the moisture content.

**Figure 5 foods-11-02009-f005:**
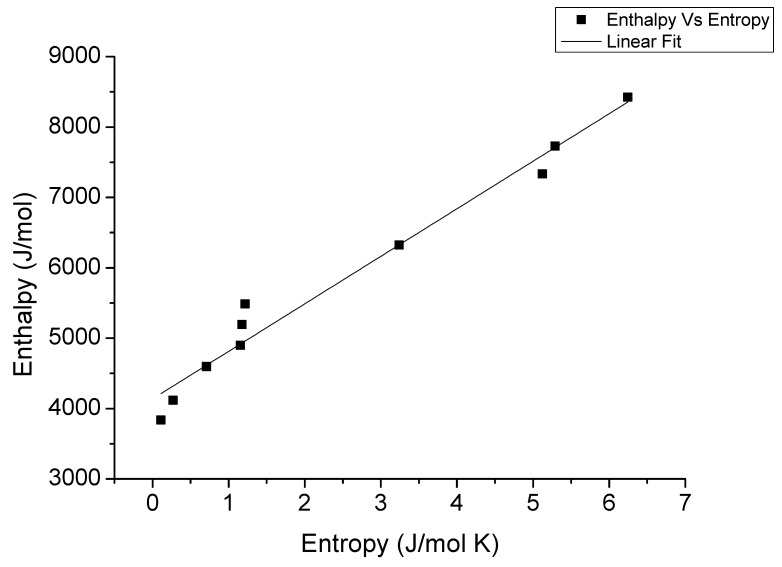
Enthalpy–entropy relationship for the desorption of pomegranate peels.

**Table 1 foods-11-02009-t001:** Mathematical models considered to describe the sorption isotherms for pomegranate peels.

Model	Equation	Parameters	Reference
GAB	$X_{w}$ $=\frac{X_{m}{\mathrm{CKa}}_{w}}{\left(1-{\mathrm{Ka}}_{w}\right)\left[1+\left(C-1\right){\mathrm{Ka}}_{w}\right]}$	X_m_: moisture content of the monolayer C: Guggenheim constant linked to the sorption heat of the monolayer K: constant linked to the sorption heat of the multilayer	[24]
BET	$X_{w}=\frac{X_{m}{\mathrm{Ca}}_{w}}{\left(1-a_{w}\right)\left[1+\left(C-1\right)a_{w}\right]}$	X_m_: moisture content of the monolayer C: papameter linked to the heat released during the sorption process	[24]
Freundlich	$X_{w}={\mathrm{Ka}}_{w}{}^{\frac{1}{n}}$	K, n: model parameters related to the product	[26]
Henderson	$X_{w}=0.01{\left[\frac{-\log\left(1-a_{w}\right)}{10^{f}}\right]}^{\frac{1}{n}}$	f, n: model parameters related to the product	[27]
Oswin	$X_{w}=A{\left[\frac{a_{w}}{1-a_{w}}\right]}^{B}$	A, B: model parameters related to the product	[15]
Peleg	$X_{w}={\mathrm{Aa}}_{w}{}^{B}+{\mathrm{Ca}}_{w}{}^{D}$	A, B, C, D: model parameters related to the product	[28]
Caurie	$X_{w}=\exp\left[a_{w}\mathrm{Ln}\left(v\right)-(\frac{1}{4.5{\;X}_{s}}\right]$	X_s_: moisture content which provides stability during storage V: parameter linked with the product	[27]
Smith	$X_{w}=B+A\;\log\left(1-a_{w}\right)$	A, B: model parameters related to the product	[29]

**Table 2 foods-11-02009-t002:** Identified parameters of the different models considered to describe the sorption isotherms of pomegranate peels and the determination coefficient (R^2^), the root-mean-squared error (RMSE) and the sum of the squared error (SSE) of the fit.

Model Name	Model Parameters and Fitting Assess	20 °C	30 °C	40 °C	50 °C
**GAB**	X_m_	0.26 ± 0.05	0.23 ± 0.04	0.20 ± 0.03	0.18 ± 0.03
	C	4.4 ± 0.8	6.0 ± 0.9	10 ± 1	12 ± 1
	K	0.9 ± 0.2	0.9 ± 0.1	1.0 ± 0.1	1.0 ± 0.1
	SSE	0.8120	0.6435	0.4523	0.5276
	R^2^	0.9554	0.9647	0.9752	0.9710
	RMSE	0.1767	0.1573	0.1319	0.1424
**BET**	X_m_	0.32 ± 0.06	0.29 ± 0.05	0.25 ± 0.04	0.23 ± 0.04
	C	0.9 ± 0.1	0.9 ± 0.1	0.9 ± 0.1	0.9 ± 0.1
	SSE	0.8530	0.7181	0.5959	0.6887
	R^2^	0.9532	0.9606	0.9673	0.9622
	RMSE	0.1777	0.1631	0.1486	0.1597
**Feundlich**	K	2.8 ± 0.6	2.9 ± 0.6	3.0 ± 0.6	3.0 ± 0.7
	N	0.18 ± 0.04	0.16 ± 0.03	0.14 ± 0.03	0.13 ± 0.03
	SSE	1.2860	1.1990	1.1740	1.2920
	R^2^	0.9294	0.9342	0.9355	0.9290
	RMSE	0.2182	0.2107	0.2085	0.2188
**Henderson**	f	−1.7 ± 0.4	−1.5 ± 0.3	−1.3 ± 0.2	−1.2 ± 0.2
	n	1.0 ± 0.2	0.9 ± 0.2	0.8 ± 0.1	0.7 ± 0.1
	SSE	1.3970	1.0860	0.7650	0.8480
	R^2^	0.9233	0.9382	0.9580	0.9534
	RMSE	0.2274	0.2005	0.1683	0.1772
**Oswin**	A	0.6 ± 0.2	0.5 ± 0.1	0.41 ± 0.08	0.39 ± 0.07
	B	0.4 ± 0.1	0.4 ± 0.1	0.50 ± 0.09	0.5 ± 0.1
	SSE	2.6620	1.8976	0.9482	0.9266
	R^2^	0.8538	0.8975	0.9479	0.9491
	RMSE	0.3140	0.2629	0.1874	0.1853
**Peleg**	A	0.47 ± 0.05	0.48 ± 0.06	0.50 ± 0.05	0.51 ± 0.06
	B	0.54 ± 0.05	0.55 ± 0.06	0.61 ± 0.06	0.63 ± 0.07
	C	2.6 ± 0.2	2.7 ± 0.3	2.8 ± 0.3	2.9 ± 0.3
	D	8.7 ± 0.8	10 ± 1	11 ± 1	12 ± 1
	SSE	0.4728	0.3348	0.2320	0.3174
	R^2^	0.9740	0.9816	0.9873	0.9826
	RMSE	0.0971	0.1157	0.0963	0.1127
**Caurie**	V	374 ± 73	648 ± 123	1352 ± 163	2398 ± 257
	X_s_	22 ± 4	24 ± 5	28 ± 5	30 ± 6
	SSE	1.039	0.9785	0.98	1.121
	R^2^	0.943	0.9463	0.9462	0.9385
	RMSE	0.1961	0.1904	0.1905	0.2037
**Smith**	A	−0.6 ± 0.1	−0.6 ± 0.1	−0.6 ± 0.1	−0.7 ± 0.1
	B	0.006 ± 0.001	−0.023 ± 0.005	−0.05 ± 0.01	−0.06 ± 0.01
	SSE	1.4020	1.1950	1.1400	1.3440
	R^2^	0.9230	0.9344	0.9374	0.9262
	RMSE	0.2279	0.2104	0.2055	0.2231

**Table 3 foods-11-02009-t003:** Thermodynamic properties of pomegranate peels at different levels of moisture content (kg H_2_O/kg d·m.).

X_eq_	q_stn_ (J/mol)	Q_st_ (J/mol)
0.11	8423.9	51,923.9
0.12	7730.0	51,230.0
0.13	7332.8	50,832.8
0.14	6322.0	49,822.0
0.15	5484.2	48,984.2
0.16	5193.3	48,693.3
0.17	4898.3	48,398.3
0.18	4596.1	48,096.1
0.19	4116.6	47,616.6
0.2	3837.7	47,337.7

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
