# Peer review of "Sorption Isotherms and Thermodynamic Properties of Pomegranate Peels"

_foods, 2022, doi:10.3390/foods11142009_

Round 1
Reviewer 1 Report
The present study aims at evaluating the thermodynamic properties of pomegranate peels at different temperatures. Water activity and moisture content were measured at different times. For the modelling of the results eight different models were considered to describe the desorption isotherms of pomegranate peel.
I believe that the paper is well written overall and the methods are carefully described. The objectives and the design of the study are clearly stated and identified in the Introduction section. Though I suggest the authors to introduce more references to the Introduction part for a better scientific soundness of the study. The authors provided the interpretation of the obtained results in a well-structured manner and through a large number of mathematical models.
On the other hand, the authors should include in the paper some statistical analysis of the obtained results. The conclusions of the study could be improved, given the large number of results obtained in the study. More references should be added in the Introduction of the research paper.
Minor English corrections:
Line 26: … of the total production
Line 35: However, whatever the use … this should be rephrased in a more scientific manner
Line 47: Replace “papers” with “scientific articles”
Author Response
Thank you very much for your valuable comments. We have carefully considered all of them and revised the manuscript accordingly. Below you can find the responses point-by-point. The recent revised contents have been marked in red in the manuscript.
The following considerations have been made:
Comment reviewer:
The present study aims at evaluating the thermodynamic properties of pomegranate peels at different temperatures. Water activity and moisture content were measured at different times. For the modelling of the results eight different models were considered to describe the desorption isotherms of pomegranate peel.
I believe that the paper is well written overall and the methods are carefully described. The objectives and the design of the study are clearly stated and identified in the Introduction section. Though I suggest the authors to introduce more references to the Introduction part for a better scientific soundness of the study. The authors provided the interpretation of the obtained results in a well-structured manner and through a large number of mathematical models.
Answer:
Following the instructions of the reviewer, more references have been added to the Introduction section.
New references in introduction section:
Saadi, W.; Rodríguez-Sánchez, S.; Ruiz, B.; Najar-Souissi, S.; Ouederni, A.; Fuente, E. From Pomegranate Peels Waste to One-Step Alkaline Carbonate Activated Carbons. Prospect as Sustainable Adsorbent for the Renewable Energy Production. J. Environ. Chem. Eng. 2022, 10, doi:10.1016/j.jece.2021.107010.
Silva, L. de O.; Garrett, R.; Monteiro, M.L.G.; Conte-Junior, C.A.; Torres, A.G. Pomegranate (Punica Granatum) Peel Fractions Obtained by Supercritical CO2 Increase Oxidative and Colour Stability of Bluefish (Pomatomus Saltatrix) Patties Treated by UV-C Irradiation. Food Chem. 2021, 362, 130159, doi:10.1016/j.foodchem.2021.130159.
El-Mesery, H.S. Improving the Thermal Efficiency and Energy Consumption of Convective Dryer Using Various Energy Sources for Tomato Drying. Alexandria Eng. J. 2022, 61, 10245–10261, doi:10.1016/j.aej.2022.03.076.
Sánchez-Torres, E.A.; Abril, B.; Benedito, J.; Bon, J.; García-Pérez, J. V. Water Desorption Isotherms of Pork Liver and Thermodynamic Properties. LWT - Food Sci. Technol. 2021, 149, doi:10.1016/j.lwt.2021.111857.
Comment reviewer:
On the other hand, the authors should include in the paper some statistical analysis of the obtained results. The conclusions of the study could be improved, given the large number of results obtained in the study. More references should be added in the Introduction of the research paper.
Answer:
As the reviewer states, the goodness of the fit of the different mathematical models considered were estimated through the determination coefficient (Eq. 1), the mean squared error (Eq. 2) and the sum of squared error (Eq. 3). These values are included in Table 2. Regarding the conclusion section, it has been changed following the instructions of the reviewer.
New version, lines 296-305:
“The sorption isotherms for pomegranate peels presented a type II shape according to the Brunauer classification. The GAB, BET and Peleg models provided the best fit for the experimental data, being useful tools for the purposes of describing the sorption isotherms of pomegranate peels. The temperature affected the experimental sorption process. This influence was also observed in the evolution with the temperature of the models parameters identified. Thus, the monolayer moisture content (Xm of GAB model), decreased as the temperature raised. The net isosteric heat of sorption was greater at lower moisture content, providing an interesting data with which to predict both the energy requirement in pomegranate peel dehydration processes and the behaviour of water molecules in sorption…”
Comment reviewer:
Minor English corrections:
Line 26: … of the total production
Answer:
Done
New version (line 31):
“…of the total production [2].”
Comment reviewer:
Line 35: However, whatever the use … this should be rephrased in a more scientific manner
Answer:
Done
New version (line 40-41):
“However, independently of the increase in value process considered, these by-products have to be stabilised to prevent degradation reactions”.
Comment reviewer:
Line 47: Replace “papers” with “scientific articles”
Answer:
Done
New version (line 54-55):
“In fact, numerous scientific articles can be found …”

Reviewer 2 Report
In manuscript “Sorption isotherms and thermodynamic properties of pomegranate peels” authors assessed the desorption isotherms of pomegranate peels at different temperatures and enthalpy, entropy and Gibbs free energy of convection drying process. Manuscript is interesting but some improvement should be made. My biggest concern goes to poor presentation of the results. Therefore my suggesting is major revision. Specific comments are listed below:
The motivation for the research should be presented in the abstract
Abstract should also include specific numerical data.
Latin words should be in italic in whole manuscript.
Line 39. Why did authors select convective drying?
Line 71. Why 60°C?
Line 76. Please corrected the writing of the indices in chemical formulas.
Line 87. What is GAB and BET?
Table 1. Authors should add references for the selected models.
Figure 1. Why were date obtained at different temperatures presented separately and not on the single chart? The size of the font on the axes is too small.
Table 2. Estimated values of model parameters should be presented with standard deviations
Decimal numbers on all graphs should be presented with decimal dots instead of decimal coma.
Author Response
Manuscript ID: foods-1794555
Sorption isotherms and thermodynamic properties of pomegranate peels
Response to Reviewer #2:
Thank you very much for your valuable comments. We have carefully considered all of them and revised the manuscript accordingly. Below you can find the responses point-by-point. The recent revised contents have been marked in red in the manuscript.
The following considerations have been made:
Comment reviewer:
In manuscript “Sorption isotherms and thermodynamic properties of pomegranate peels” authors assessed the desorption isotherms of pomegranate peels at different temperatures and enthalpy, entropy and Gibbs free energy of convection drying process. Manuscript is interesting but some improvement should be made. My biggest concern goes to poor presentation of the results. Therefore my suggesting is major revision. Specific comments are listed below:
The motivation for the research should be presented in the abstract
Answer:
The motivation of the paper has been introduced in the abstract
New version (line 10-14):
“Convective drying is the most used technique to stabilize food industry by-products, which permit a later processing. A thorough knowledge of the relationship between moisture content a water activity permits the optimization of not only drying operation but the setting of storage conditions. Thus, the thermodynamic properties of pomegranate peels were determined during the desorption process.”
Comment reviewer:
Abstract should also include specific numerical data.
Answer:
Specific numerical data has been included in the abstract
New version (line 16-17):
“the theoretical GAB model and the empirical Peleg model were the ones that achieved the best fit (R2 of 0.9554 and 0.974, respectively)”
New version (line 18-20):
“The isosteric heat determined from the sorption isotherms decreased regularly as the equilibrium moisture content rose (from 8423.9 J/mol at 0.11 kgH2O/kg d.m. to 3837.7 J/mol at 0.2 kgH2O/kg d.m.).”
Comment reviewer:
Latin words should be in italic in whole manuscript.
Answer:
Done
Comment reviewer:
Line 39. Why did authors select convective drying?
Answer:
Convective drying is, in this moment, the most used method to stabilise food products and by-products because its effectiveness and the simplicity to implement. Moreover, the reduction of water content makes also simpler and cheaper the transport and storage of the dried products. Changes has been introduced in the text to highlight these characteristics.
New version (line 42-46):
“In this sense, convective drying is one the most common food preservation methods. This drying technique is characterized by its simplicity and reliability. The main purpose of drying is to obtain products with reduced water activity (aw), which limits the undesirable degradation of interesting compounds thereby prolonging the shelf life. The moisture content reduction also makes easier the later transport and storage operations.”
Comment reviewer:
Line 71. Why 60°C?
Answer:
This moderate temperature is widely use to dry food products and by-products because means a compromise solution with a reasonable velocity of the process and the preservation of natural characteristics of the product. Changes has been introduced to clarify this point.
New version (line 77-81):
“For this purpose, peel pomegranate samples were placed into a sample holder and then into a laboratory convective oven at 60°C. This temperature is widely used for the drying of food products and by-products because represents a compromise solution between drying kinetics and the preservation of interesting compounds [22]”
New reference
[22] Borsini, A.A.; Llavata, B.; Umaña, M.; Cárcel, J.A. Artichoke by Products as a Source of Antioxidant and Fiber. How It Can Be Affected by Drying Temperature. Foods 2021, 10, 459. https://doi.org/10.3390/ foods10020459
Comment reviewer:
Line 76. Please corrected the writing of the indices in chemical formulas.
Answer:
Done
New version (line 86-87):
“The hygrometer was previously calibrated using saturated solutions of different salts which covered the whole range of water activity (LiCl, MgCl2, Mg(NO3)2, NaCl, BaCl2 and K2Cr2O7).”
Comment reviewer:
Line 87. What is GAB and BET?
Answer:
Both are theoretical models to describe isotherms, and their names correspond with the acronym derived of the authors names. Thus, GAB is the acronym of Guggenheim-Anderson-de Boer and BET of Brunauer-Emmett-Teller. Both have been introduced in the text to avoid misunderstandings
New version (line 97-98):
“…two theoretical models, GAB (Guggenheim-Anderson-de Boer) and BET (Brunauer-Emmett-Teller),…”
Comment reviewer:
Table 1. Authors should add references for the selected models.
Answer:
References of each model have been added to the Table 1
Comment reviewer:
Figure 1. Why were date obtained at different temperatures presented separately and not on the single chart? The size of the font on the axes is too small.
Answer:
The goal of presenting the isotherms obtained at different temperature in separate charts was double, to show in a clearly way the trend of the different experimental isotherms and to show the adequate fit of Peleg’s model in every case. Regarding the influence of temperature in the isotherms, it was quantified by the modelling of experimental data. Thus, model parameters from both, theoretical models, such as GAB’s model, and empirical ones, such as Peleg’s model, showed these differences.
As for the size of the font on the axes, it has been increased in the new version following the instructions of the reviewer, not only in figure 1 but also in figure 2, 3 and 4.
Comment reviewer:
Table 2. Estimated values of model parameters should be presented with standard deviations
Answer:
The fit of the different models was carried out using in a single run all the pairs of experimental data of moisture content-water activity available for each temperature, including the replicates. The goodness of the fit were estimated by the estimation of three statistics, the determination coefficient (Eq. 1), the mean squared error (Eq. 2) and the sum of squared error (Eq. 3). These values are included in Table 2 and permit to quantify the variability of the experimental data explained by each model.
Comment reviewer:
Decimal numbers on all graphs should be presented with decimal dots instead of decimal coma.
Answer:
Thank you for the observation. Every decimal coma has been replaced by decimal dots

Round 2
Reviewer 2 Report
The authors answered most of my comments, but they should include standard deviations, errors, or coefficient intervals in Table 2. I agree with the criteria for model goodness of fit estimation, but my comment was on the estimation of kinetic parameter values. The precision of the parameter estimates must be known and presented in the manuscript. Therefore, my suggestion is a minor revision according to the mentioned comment.
Author Response
Comment reviewer:
The authors answered most of my comments, but they should include standard deviations, errors, or coefficient intervals in Table 2. I agree with the criteria for model goodness of fit estimation, but my comment was on the estimation of kinetic parameter values. The precision of the parameter estimates must be known and presented in the manuscript. Therefore, my suggestion is a minor revision according to the mentioned comment.
Answer:
Following the advice of the reviewer, the intervals of confidence of the model’s parameters have been included in Table 2
